# Resistance to post-emergent herbicides is becoming common for grass weeds on New Zealand wheat and barley farms

Christopher E. Buddenhagen[1]*, Trevor K. James[1], Zachary Ngow[1], Deborah L. Hackell[1], M. Phil Rolston[2], Richard J. Chynoweth[2], Matilda Gunnarsson[2], Fengshuo Li[3,4], Kerry C. Harrington[4], Hossein Ghanizadeh[4]*

1 AgResearch Ltd., Hamilton, New Zealand, 2 Foundation for Arable Research, Christchurch, New Zealand, 3 College of Horticulture and Landscape Architecture, Northeast Agricultural University, Harbin, China, 4 School of Agriculture and Environment, Massey University, Palmerston North, New Zealand

* Chris.Buddenhagen@agresearch.co.nz (CEB); h.ghanizadeh@massey.ac.nz (HG)

**Data Availability Statement:** All relevant data are within the paper and its Supporting information files.

## Abstract

To estimate the prevalence of herbicide-resistant weeds, 87 wheat and barley farms were randomly surveyed in the Canterbury region of New Zealand. Over 600 weed seed samples from up to 10 mother plants per taxon depending on abundance, were collected immediately prior to harvest (two fields per farm). Some samples provided by agronomists were tested on an ad-hoc basis. Over 40,000 seedlings were grown to the 2–4 leaf stage in glasshouse conditions and sprayed with high priority herbicides for grasses from the three modes-of-action acetyl-CoA carboxylase (ACCase)-inhibitors haloxyfop, fenoxaprop, clodinafop, pinoxaden, clethodim, acetolactate synthase (ALS)-inhibitors iodosulfuron, pyroxsulam, nicosulfuron, and the 5-enolpyruvyl shikimate 3-phosphate synthase (EPSPS)-inhibitor glyphosate. The highest manufacturer recommended label rates were applied for the products registered for use in New Zealand, often higher than the discriminatory rates used in studies elsewhere. Published studies of resistance were rare in New Zealand but we found weeds survived herbicide applications on 42 of the 87 (48%) randomly surveyed farms, while susceptible reference populations died. Resistance was found for ALS-inhibitors on 35 farms (40%) and to ACCase-inhibitors on 20 (23%) farms. The number of farms with resistant weeds (denominator is 87 farms) are reported for ACCase-inhibitors, ALS-inhibitors, and glyphosate respectively as: *Avena fatua* (9%, 1%, 0% of farms), *Bromus catharticus* (0%, 2%, 0%), *Lolium* spp. (17%, 28%, 0%), *Phalaris minor* (1%, 6%, 0%), and *Vulpia bromoides* (0%, not tested, 0%). Not all farms had the weeds present, five had no obvious weeds prior to harvest. This survey revealed New Zealand's first documented cases of resistance in *P. minor* (fenoxaprop, clodinafop, iodosulfuron) and *B. catharticus* (pyroxsulam). Twelve of the 87 randomly sampled farms (14%) had ALS-inhibitor chlorsulfuron-resistant sow thistles, mostly *Sonchus asper* but also *S. oleraceus*. Resistance was confirmed in industry-supplied samples of the grasses *Digitaria sanguinalis* (nicosulfuron, two maize farms), *P. minor* (iodosulfuron, one farm), and *Lolium* spp. (cases included glyphosate, haloxyfop, pinoxaden, iodosulfuron, and pyroxsulam, 9 farms). Industry also supplied

**Funding:** All the authors worked under the Ministry of Business, Innovation and Employment [grant number C10X1806] to AgResearch Ltd.: "Improved weed control and vegetation management to minimize future herbicide resistance." AgResearch Ltd is a crown (i.e., the Government) owned research institution in New Zealand. https://www.mbie.govt.nz/assets/e5a12d67b2/2018-endeavour-round-successful-projects.pdf The funders had no role in study design, data collection and analysis, decision to publish, or preparation of the manuscript.

**Competing interests:** The authors have declared that no competing interests exist.

*Stellaria media* samples that were resistant to chlorsulfuron and flumetsulam (ALS-inhibitors) sourced from clover and ryegrass fields from the North and South Island.

## Introduction

Weed control programs that use herbicides have proven to be cost-effective for improving yields of staple crops by an average of 30% [1], and typically provide a 2-4-fold economic return [2]. They are also a key element in no-till planting programs for wheat (*Triticum aestivum* L.) and barley (*Hordeum vulgare* L.) farms in New Zealand that improve soil structure and prevent soil loss through erosion [3]. Nevertheless, farmer practices worldwide have led to the selection of weeds with infrequent genetic mutations that confer resistance to the herbicides, allowing weeds to escape control, reproduce and form resistant populations [4]. Globally, herbicide resistance is common in arable crops such as wheat (344 cases and 83 species) and barley (87 cases, 47 species) [5]. Based on worldwide patterns of resistance, Ngow et al [6] identified 16 species with a high risk of developing resistance in New Zealand wheat and barley fields (eight were grasses *Avena fatua* L., *A. sterilis* L., *Digitaria sanguinalis* (L.) Scop., *Echinochloa crus-galli* (L.) P. Beauv., *Lolium multiflorum* Lam., *L. perenne* L., *Phalaris minor* Retz., and *Poa annua* L.; these grasses were in the top 10 for risk of developing herbicide resistance).

In any given year >50% of New Zealand arable production areas are under wheat (~45000 ha) and barley (~55000 ha) rotations [7] and only a small proportion of the more than 800 farms (<50 certified farms) are registered as organic [8]. Under intense management, production levels are high, with farmers in New Zealand obtaining world record yields of wheat (17.39 tons/ha) and barley (13.8 tons/ha) in 2017 and 2015, respectively [9,10]. Yet only a few instances of herbicide resistance in ryegrass species *L. perenne* and *L. multiflorum* have been documented to date in wheat and barley in New Zealand [11], and in *A. fatua* [12]. There is at least one case of *Stellaria media* (L.) Vill. resistance recorded in an oat crop [13]. Compared to Australia or the United States, there appears to be only a small number of resistance cases documented in New Zealand arable farms [14]. Frequent crop rotations may allow New Zealand farmers to rotate herbicidal modes-of-action (e.g., effective against broadleaf or grass weeds), implement resting periods, stale seed bed or cultivation steps; these are widely regarded as key elements in best practice for resistance management [15,16]. Species commonly included in wheat and barley rotations in New Zealand are pasture, spring-sown peas (*Pisum sativum* L.), linseed (*Linum usitatissimum* L.), ryegrass, clover (*Trifolium repens* L.), oilseed rape (*Brassica napus* L.), and wheat or barley. The higher manufacturer label recommended application rates in New Zealand [17] (compared to Australia or the USA) could also have an influence on the rates of resistance development and detection (discussed later). Another plausible explanation for the low number of resistance cases in wheat and barley farms in New Zealand is that the problem is simply under-investigated.

This study aimed to determine the prevalence of herbicide-resistant weeds on arable farms with wheat and barley rotations in a northern (near Lincoln) and southern locality (near Timaru) of the Canterbury region in the South Island of New Zealand. Surveys focused on randomly selected farms and sampled weeds with mature seeds immediately before to crop harvest. As grass weeds were the most common, we focus our reporting on those (but mention other cases). This work represents the first random survey to detect herbicide resistance for any agricultural sector in New Zealand.

## Materials and methods

### Collection of plant material

Weeds seeds were collected from 87 randomly selected arable farms from the Foundation for Arable Research (FAR) member database in January and February 2019 and 2020. This represents 21% of the possible farms in the selected regions. In 2019, 52 farms were surveyed between the Rakaia and Waimakariri Rivers near Lincoln, and in 2020, 35 farms near Timaru were visited. The FAR member database is thought to contain at least 90% of arable farmers in New Zealand. Seeds were collected from one or two fields per farm, usually with wheat or barley, or more rarely clover seed crops. If weeds were present and depending on abundance, seed samples from up to 10 individual weeds with viable seeds were collected for each weed species (grass weeds were the most common and the focus of this study). We focused on detecting the presence of resistant plants of any weed detected at the level of farms, not on within farm population level differences. Our sampling rates are more suited to the reliable detection of outcrossing species e.g., *Lolium* [18,19], but the presence of each weed species within farms varied stochastically, and time and resource considerations came into play. Combined with our focus on just two fields, we accepted that our estimates of resistance prevalence in farms would be conservative (lower than the true rate). If weeds for a species were frequent in a field, an effort was made to space out the collections from across the whole field. Plants growing in mid-field (as opposed to edges) were favoured. In 2019, seed from each species was collected and bulked together for a field sample. *Lolium multiflorum* and *L. perenne* seed was separated based on field determinations of species (based on awn length and leaf blade width). However, most ryegrass seed samples were found to be difficult to distinguish, and many were hybrids. In 2020, 35 farms, primarily from Southern Canterbury centred around Timaru, were surveyed. Unlike 2019, in 2020 we kept separate seed samples for each mother plant. Seed samples were labelled with location and species information and stored in paper envelopes or bags and kept in a cool store at 4˚C until planting. A single georeferenced point was recorded for each field sampled. For this paper, we focus our results on the grasses and comment on our results for a few other cases, including some detected via ad-hoc industry supplied samples.

Susceptible controls for *A. fatua*, *B. diandrus* Roth and *B. catharticus* Vahl were sourced from an organic farm near Methven. Susceptible *Lolium* spp. samples used in this study is the same as that described in an earlier New Zealand herbicide resistance study, diploid varieties Trojan and Tabu for *L. perenne* and *L. multiflorum*, respectively [20]. Known susceptible controls for *Vulpia bromoides* L. Gray and *P. minor* were not available at the time of herbicide treatment, but some samples in every treatment block did show 100% mortality. Susceptible *Sonchus asper* (L.) Hill, *S. oleraceus* L. and *S. media* were sourced from pastures near Ruakura.

After advising that our herbicide resistance project provided free testing, industry representatives and agronomists sent us seeds from several suspected resistant plants i.e., that were not part of the random survey. We tested ryegrass from additional 11 farms with suspected resistance to a variety of herbicides, including pyroxsulam, pinoxaden and one case of glyphosate resistance (mostly from wheat and barley fields). Additionally, *D. sanguinalis* with suspected resistance to nicosulfuron (ALS inhibitor) from a maize crop from the Waikato region of the North Island. *A. fatua* from three farms in Canterbury were suspected of pinoxaden and pyroxsulam resistance (acetyl-CoA carboxylase (ACCase)-inhibitor and ALS-inhibitor, respectively). A sample of *P. minor* suspected of being resistant to ALS-inhibitors was also provided for testing (also from Canterbury). We were also supplied with two samples of *S. media* suspected of resistance to flumetsulam (acetolactate synthase (ALS)-inhibitor). One sample from barley in the South Island and one from ryegrass seed crops in the North Island.

## Growing plants and spraying

**Plants grown at Ruakura.**   At the Ruakura Research Centre, Hamilton, we planted 10–40 (usually 30) weed seeds per pot in March 2019 from each sample in three to six 9 cm x 9 cm x 9 cm black plastic pots (one per herbicide that we tested) containing sterile, commercial potting mix (Daltons) that included a slow-release fertilizer. A susceptible control was also grown. Pots were kept moist (watered every 2–3 days) and kept in a temperature regulated glasshouse at Ruakura and maintained at between 18 and 25°C. In April and May 2020, we changed our protocol. Twenty seeds were sown into propagation trays (22 cm x 35 cm x 5 cm) into one of six rows or lanes, such that each tray could contain four to five field collected samples and a known herbicide susceptible control and a known herbicide resistant control, if available. Again, each sample would have its seed spread between three to six propagation trays, with one tray per herbicide tested. Samples in these trays were kept moist (watered every 2–3 days) and kept in a temperature regulated glasshouse at Ruakura and maintained at between 18 and 25°C. Propagation trays with *Lolium* spp. seed were planted at 2–3 mm depth, watered and chilled in a cool store at 4°C for 48 hours before being placed in the glasshouse. *A. fatua* samples from 2020 were dehusked and soaked in 0.1% $KNO_3$ for 24 hours before planting into trays at 3–4 mm depth [e.g., 21]. Grass seedlings were raised to the 2–4 leaf stage before herbicide was applied. Before herbicide treatment, seedlings in all pots or propagation trays were counted.

Depending on the availability of adequate seed in a sample, the number of herbicides that could be tested changed. Herbicides were tried in the priority order shown for each taxon (Table 1). For example, a sample of 60 seeds would only be tested against the top three to five priority herbicides (Table 1) as we tried to maintain between 10 and 25 seeds per treatment. The same priority order was used for samples treated at Massey University (see below). All herbicide treatments were applied using the highest recommended label rate for the herbicide being tested (Table 1) with a moving belt sprayer using a single TeeJet TT11002 fan nozzle at 200 kPa, positioned 440 mm above the top of the pots/trays to apply 200 L/ha. Glyphosate was tested because it is commonly used prior to planting for seed bed preparation. We included isoproturon on *V. bromoides* which is a photosystem II inhibitor because industry consultants thought it is effective, even though this species is not mentioned on the herbicide label. In 2019, up to 12 pots were grouped into trays (22cm x 35cm x 5 cm) nursery for spraying. Up to about 22 nursery trays per herbicide treatment could be sprayed with a single herbicide at any given time due to the 1 L capacity of the spray tank reservoir. Only one or two susceptible controls (in individual pots) were used per herbicide treatment. In 2020, as mentioned above seeds were planted out in lanes across each tray with a susceptible control in one of the lanes per propagation tray. *Sonchus asper* and *S. oleraceus* from the random surveys were treated with chlorsulfuron 20 g ai/ha (AgPro Chloro®) with a non-ionic surfactant (0.1%).

We also report results from a few other weeds supplied to us by industry agronomists that were tested. *S. media* samples were sourced from ryegrass rotation (near Matamata on the North Island) and one sample from Ashburton in the South Island from a field planted in clover and ryegrass were treated with flumetsulam 30 g ai/ha (Preside®) and a paraffinic oil surfactant (0.5%), and chlorsulfuron 20 g ai/ha (AgPro Chloro®) with a non-ionic surfactant (0.25%). *D. sanguinalis* supplied to us from two maize (*Zea mays* L.) farms near Matamata on the North Island was treated with nicosulfuron 60 g ai/ha with paraffinic oil surfactant (0.5%) (S1 Appendix).

**Plants grown at Massey University.**   *Avena fatua* seed samples from 2019 were processed at Massey University, Palmerston North. To overcome seed dormancy, seed samples were dehusked and soaked in 800 ppm gibberellic acid ($GA_3$) overnight at room temperature before

**Table 1. Herbicides and application rates for the grass weed species.**

| Weed | Priority Order | Trade Name | Active Ingredients | Rate g ai per ha | Adjuvant | Adjuvant rate |
|---|---|---|---|---|---|---|
| *Lolium perenne* | 1 | Ignite | haloxyfop | 250 | none | none |
| *Lolium perenne* | 2 | Twinax | pinoxaden | 30 | Adigor 440 g/L methyl esters of canola oil, fatty acids solvent, 222 g/L liquid hydrocarbons | 0.50% |
| *Lolium perenne* | 3 | Rexade | halauxifen-methyl and pyroxsulam | 15 | Actiwett 950 g/litre linear alcohol ethoxylate | 0.25% |
| *Lolium perenne* | 4 | Weedmaster | glyphosate | 1458 | Pulse 800 g/litre organosilicone modified polydimethy siloxane | 0.10% |
| *Lolium perenne* | 5 | Hussar | iodosulfuron | 7.5 | Partner vegetable oil polymer | 0.50% |
| *Lolium perenne* | 6 | Sequence | clethodim | 120 | Bonza 471 g/L paraffin oil | 0.50% |
| *Lolium multiflorum* | 1 | Ignite | haloxyfop | 125 | none | |
| *Lolium multiflorum* | 2 | Twinax | pinoxaden | 30 | Adigor | 0.50% |
| | 3 | Simplicity | pyroxsulam | 15 | Actiwett | 0.25% |
| *Lolium multiflorum* | 3 | Rexade | halauxifen-methyl and pyroxsulam | 5/10 and 15/30 | Actiwett | 0.25% |
| *Lolium multiflorum* | 4 | Weedmaster | glyphosate | 1458 | Pulse | 0.10% |
| *Lolium multiflorum* | 5 | Hussar | iodosulfuron | 7.5 | Partner | 0.50% |
| *Lolium multiflorum* | 6 | Sequence | clethodim | 120 | Bonza | 0.50% |
| *Avena fatua* | 1 | Puma-S | fenoxaprop | 51.75 | none | |
| *Avena fatua* | 2 | Twinax | pinoxaden | 25 | Adigor | 0.50% |
| *Avena fatua* | 3 | Sequence | clethodim | 120 | Bonza | 0.50% |
| *Avena fatua* | 4 | Simplicity | pyroxsulam | 15 | Actiwett | 0.25% |
| *Avena fatua* | 4 | Rexade | halauxifen-methyl and pyroxsulam | 5/10 and 15/30 | Actiwett | 0.25% |
| *Avena fatua* | 5 | Weedmaster | glyphosate | 702 | Pulse | 0.10% |
| *Bromus catharticus* | 1 | Ignite | haloxyfop | 250 | none | none |
| *Bromus catharticus* | 2 | Sequence | clethodim | 240 | Bonza | 0.50% |
| *Bromus catharticus* | 3 | Rexade | halauxifen-methyl and pyroxsulam | 5/10 and 15/30 | Contact 980 g/litre linear alcohol ethoxylate. | 0.25% |
| *Bromus catharticus* | 4 | Weedmaster | glyphosate | 540 | Pulse | 0.10% |
| *Bromus diandrus* | 1 | Ignite | haloxyfop | 250 | none | |
| *Bromus diandrus* | 2 | Sequence | clethodim | 240 | Bonza | 0.50% |
| *Bromus diandrus* | 3 | Simplicity | pyroxsulam | 15 | Contact | 0.25% |
| *Bromus diandrus* | 3 | Rexade | halauxifen-methyl and pyroxsulam | 5/10 and 15/30 | Actiwett | 0.25% |
| *Bromus diandrus* | 4 | Weedmaster | glyphosate | 540 | Pulse | 0.10% |
| *Bromus hordeaceus* | 1 | Ignite | haloxyfop | 250 | none | |

*(Continued)*

**Table 1.** (Continued)

| Weed | Priority Order | Trade Name | Active Ingredients | Rate g ai per ha | Adjuvant | Adjuvant rate |
|------|---------------|------------|--------------------|-----------------|----------|---------------|
| *Bromus hordeaceus* | 2 | Sequence | clethodim | 240 | Bonza | 0.50% |
| *Bromus hordeaceus* | 3 | Simplicity | pyroxsulam | 15/30 | Contact | 0.25% |
| *Bromus hordeaceus* | 3 | Rexade | halauxifen-methyl and pyroxsulam | 5/10 and 15/30 | Contact | 0.25% |
| *Bromus hordeaceus* | 4 | Weedmaster | glyphosate | 540 | Pulse | 0.10% |
| *Phalaris minor* | 1 | Hussar | iodosulfuron | 7.5 | Partner | 0.50% |
| *Phalaris minor* | 2 | Mandate | clodinafop | 24 | Uptake 582 g/litre paraffinic oils and 240 g/litre alkoxylated alcohol non-ionic surfactants | 0.50% |
| *Phalaris minor* | 3 | Sequence | clethodim | 240 | Bonza | 0.50% |
| *Phalaris minor* | 4 | Weedmaster | glyphosate | 702 | Pulse | 0.10% |
| *Phalaris minor* | 5 | Ignite | haloxyfop | 60 | none | |
| *Phalaris minor* | 6 | Twinax | pinoxaden | 30 | Adigor | 0.50% |

Herbicides were tried in the priority order shown for each taxon (see Methods for detail). Seeds of *A. fatua* and *Bromus* spp. were replanted and sprayed at the 30 g ai/ha rate for pyroxsulam (Rexade GoDri). Rates used for *S. asper*, *S. oleraceus*, *S. media* and *D. sanguinalis* are reported in the text. Adjuvant ingredients are described at first mention.

they were chilled for 3 days at 5 ˚C. The seeds were then planted into polyethylene planter bags (PB2, 1.2 L) containing potting mix and a slow-release fertilizer as described by Ghanizadeh and Harrington [22]. There were three replicates for each population and herbicide combination. Each replicate consisted of 10–17 seeds planted in one pot for most of the populations and herbicide combinations. However, due to limited seeds, for a few populations, there were seven to 10 seeds per replicate. The pots were kept in a glasshouse with a capillary irrigation system. The minimum/maximum daily temperature in the glasshouse was 19.6/22.3 ˚C and the average relative humidity was 55%. At 4–5 days after planting, 100% emergence was recorded for all replicates. At 10 days after emergence, when the plants were at the 2-leaf stage, they were treated with herbicides. Each herbicide was applied using a dual-nozzle (Teejet 730231 flat-fan nozzles) laboratory track sprayer calibrated to deliver 230 L/ha of herbicide solution at 200 kPa.

**Determining mortality.** Mortality was assessed 2 weeks after spraying for the shorter acting herbicides (e.g., glyphosate and haloxyfop) and after 3 weeks for longer acting herbicides (e.g., pyroxsulam, iodosulfuron, clethodim). If plant stem tissue near the base was soft and discoloured or if most of the plant was brown or black, then the plant was determined to be dead. In all herbicide resistance cases reported here the susceptible control died if it were available, or some of the samples sprayed at the same time experienced 100% mortality if not, i.e., *Phalaris*, and *Vulpia*.

## Statistical analysis

Maps and pivot tables statistics were produced in the R statistical platform using the ggmap and tidyverse packages [23–25] this included terrain map tiles which contains information from OpenStreetMap contributors and OpenStreetMap Foundation, which is made available under the Open Database License [26]. We were conservative about determining a resistance

case, we excluded samples where germination was poor (<5 seedlings). A farm was designated as "resistant" if a sample from it had more than 10% of plants survive a treatment and number that survived was greater than three plants. This threshold supported the effort to determine what proportion of farms are likely to have resistance. The 95% confidence interval for the proportion of farms with resistance was estimated using the R function binom.test. The prop.test function in R was used to test the hypothesis that a higher proportion of the tested Timaru farms (thought to have less crop rotation) had cases of resistance compared to Lincoln [25].

## Results

The results from over 600 samples of grasses collected from two sampling regions are reported, including a total of 87 farms that were near mid-northern Canterbury near Lincoln (52 farms) and southern Canterbury near Timaru (35 farms). Each farm would often have more than one crop type, sampled fields included wheat (61 farms), barley (30 farms) and white clover (21), but we included <2 farm fields with linseed, beets (*Beta vulgaris* L.), and peas. The following common weedy grasses were found as survivors prior to harvest on a sizeable number of the 87 farms we visited: *A. fatua* (52 farms), *B. diandrus* (16), *B. hordeaceus* (19), *B. catharticus* (29), *Lolium* spp. for suspected hybrids (23), *L. multiflorum* (38), *L. perenne* (46), *Lolium* spp. (occurred on a total 57 farms), *P. minor* (17) and *V. bromoides* (14). Only three broadleaf weeds were common, *S. asper* and *S. oleraceus* were found on 27 farms and *Achillea millefolium* L. (yarrow) on five. All other weeds were collected from three or fewer farms. Some seed samples had poor germination and were not included in our tests of resistance. Results are presented for the common grasses surviving label rate applications of different post-emergent herbicides but grouped by weed genus and herbicide modes-of-action. More detailed results broken down by farm and herbicide active ingredients are provided in the supplementary materials (S1 Appendix). Farms with plants (within each weed genus) surviving treatment with one or more herbicides for a mode-of-action are indicated in Table 2. A spatial presentation of the same data indicates where the farms with resistance were located (Fig 1), and resistance to herbicides in the given mode-of-action (Fig 2).

In the random survey, some form of resistance (for any taxon) was detected on 42 farms (48% of those surveyed, with a 95% confidence interval of 37%-59%), resistance was found for ALS-inhibitors on 35 farms (40%) and to ACCase-inhibitors on 20 (23%) farms. No cases of glyphosate-resistant grasses were detected in the random survey. Only *V. bromoides* was tested for the photosystem II (PSII) inhibiting herbicide isoproturon and no resistance was detected. To strengthen our determination of resistance to a particular herbicide mode-of-action, multiple herbicides were tested for some modes-of-action depending on the species, ACCase-inhibitors (haloxyfop, clodinafop, fenoxaprop, pinoxaden, clethodim) and two herbicides in the ALS-inhibitors (iodosulfuron and pyroxsulam) were tested. In addition, different weeds were subjected to one or more of these herbicides, in priority order according to the quantity of seed available (Table 1). *Avena fatua* seedlings were often not killed by the application of the ALS-inhibiting herbicide pyroxsulam at the 15 g ai/ha rate but only one farm was confirmed to have resistant plants when sprayed at 30 g ai/ha. While eight (9%) farms had *A. fatua* resistant to ACCase-inhibiting herbicides, samples from six farms survived fenoxaprop but were killed when treated with haloxyfop or clethodim (S1 Appendix). In the case of *Lolium* species, we found that it was hard to get an accurate taxonomic identification from the seed samples and doubts about their provenance persisted even though most surviving plants were grown on to flowering stage. Many ryegrass hybrids are grown in New Zealand, and we attempted to class them based on awn length, mid-rib prominence and lamina width. Nevertheless, we used the higher herbicide rates recommended for *L. perenne* in the second year of the study (for all

**Table 2. The number of farms with herbicide resistant grass weeds sourced from 87 randomly surveyed wheat and barley farms near Lincoln (52 farms) and Timaru (35 farms) in the South Island of New Zealand.**

| Genus | Site of action | Source | Resistant Farms | Tested Farms | Farms with weed | % Resistant of tested | % Resistant of surveyed farms |
|---|---|---|---|---|---|---|---|
| *Avena* | ACCase | Lincoln | 5 | 23 | 29 | 22 | 10 |
| *Avena* | ACCase | Timaru | 3 | 14 | 23 | 21 | 9 |
| *Avena* | ALS | Lincoln | 0 | 5 | 29 | 0 | 0 |
| *Avena* | ALS | Timaru | 1 | 9 | 23 | 11 | 3 |
| *Avena* | EPSPS | Lincoln | 0 | 8 | 29 | 0 | 0 |
| *Avena* | EPSPS | Timaru | 0 | 8 | 23 | 0 | 0 |
| *Bromus* | ACCase | Lincoln | 0 | 21 | 21 | 0 | 0 |
| *Bromus* | ACCase | Timaru | 0 | 19 | 24 | 0 | 0 |
| *Bromus* | ALS | Lincoln | 0 | 18 | 21 | 0 | 0 |
| *Bromus* | ALS | Timaru | 2 | 20 | 24 | 10 | 6 |
| *Bromus* | EPSPS | Lincoln | 0 | 21 | 21 | 0 | 0 |
| *Bromus* | EPSPS | Timaru | 0 | 14 | 24 | 0 | 0 |
| *Lolium* | ACCase | Lincoln | 7 | 28 | 30 | 25 | 13 |
| *Lolium* | ACCase | Timaru | 8 | 18 | 26 | 44 | 23 |
| *Lolium* | ALS | Lincoln | 12 | 28 | 30 | 43 | 23 |
| *Lolium* | ALS | Timaru | 12 | 18 | 26 | 67 | 34 |
| *Lolium* | EPSPS | Lincoln | 0 | 27 | 30 | 0 | 0 |
| *Lolium* | EPSPS | Timaru | 0 | 18 | 26 | 0 | 0 |
| *Phalaris* | ACCase | Lincoln | 1 | 10 | 10 | 10 | 2 |
| *Phalaris* | ACCase | Timaru | 0 | 5 | 7 | 0 | 0 |
| *Phalaris* | ALS | Lincoln | 2 | 9 | 10 | 22 | 4 |
| *Phalaris* | ALS | Timaru | 3 | 6 | 7 | 50 | 9 |
| *Phalaris* | EPSPS | Lincoln | 0 | 6 | 10 | 0 | 0 |
| *Phalaris* | EPSPS | Timaru | 0 | 4 | 7 | 0 | 0 |
| *Vulpia* | ACCase | Lincoln | 0 | 3 | 6 | 0 | 0 |
| *Vulpia* | ACCase | Timaru | 0 | 8 | 8 | 0 | 0 |
| *Vulpia* | EPSPS | Lincoln | 0 | 6 | 6 | 0 | 0 |
| *Vulpia* | EPSPS | Timaru | 0 | 7 | 8 | 0 | 0 |
| *Vulpia* | PSII | Lincoln | 0 | 3 | 6 | 0 | 0 |
| *Vulpia* | PSII | Timaru | 0 | 6 | 8 | 0 | 0 |
| *Avena* | ACCase | Industry | 1 | 3 | 3 | 33 | NA |
| *Avena* | ALS | Industry | 0 | 2 | 2 | 0 | NA |
| *Avena* | EPSPS | Industry | 0 | 1 | 1 | 0 | NA |
| *Bromus* | ALS | Industry | 0 | 1 | 1 | 0 | NA |
| *Digitaria* | ALS | Industry | 3 | 3 | 3 | 100 | NA |
| *Lolium* | ACCase | Industry | 7 | 11 | 11 | 64 | NA |
| *Lolium* | ALS | Industry | 8 | 11 | 11 | 73 | NA |
| *Lolium* | EPSPS | Industry | 1 | 9 | 9 | 11 | NA |
| *Phalaris* | ACCase | Industry | 0 | 1 | 1 | 0 | NA |
| *Phalaris* | ALS | Industry | 1 | 1 | 1 | 100 | NA |
| *Phalaris* | EPSPS | Industry | 0 | 1 | 1 | 0 | NA |

Industry provided samples of suspected resistant plants are also reported but we do not include their region. We report pyroxsulam resistance levels for oats and bromes treated at 30 g/ai ha. Not all farms with surviving weeds sampled could be tested because of germination problems. ACCase = acetyl CoA carboxylase, ALS = acetolactate synthase, EPSPS = 5-enolpyruvylshikimate-3-phosphate synthase and PS II = photosystem II.

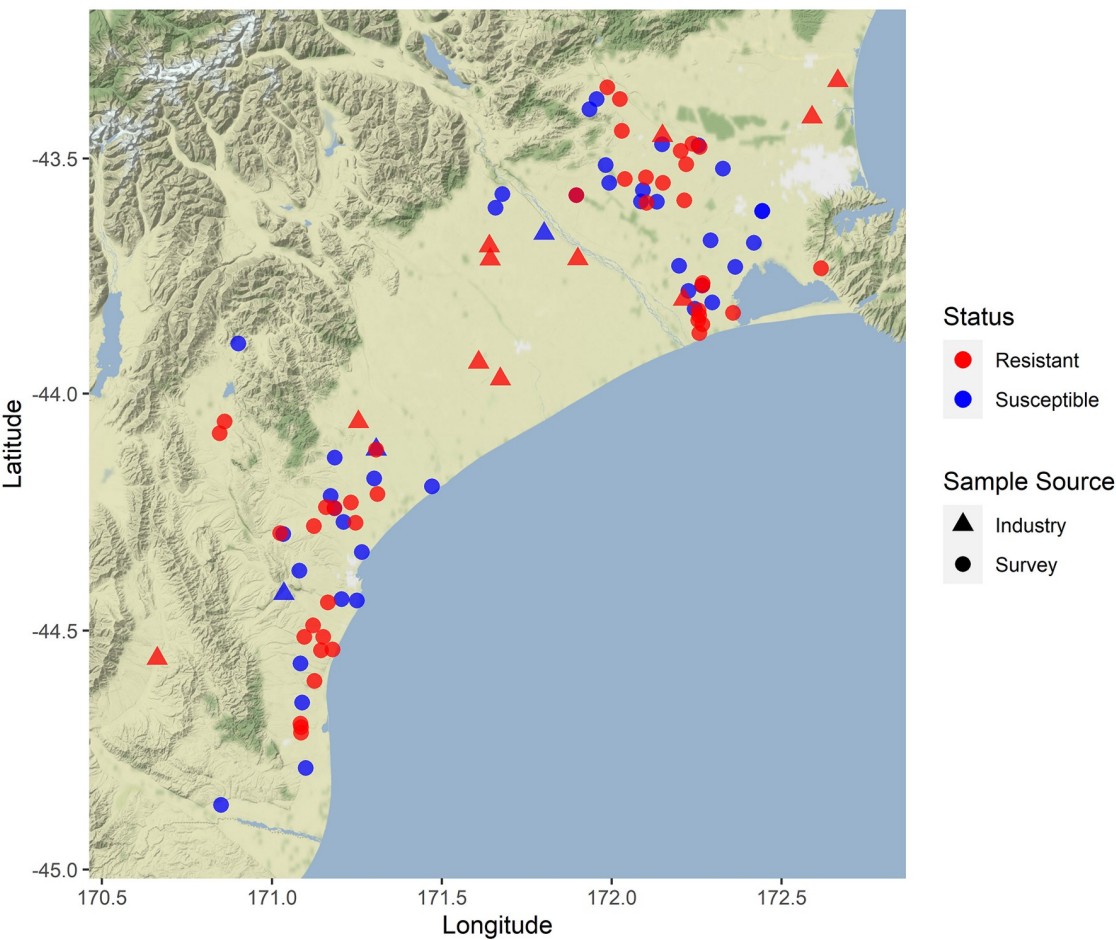

**Fig 1. Map of the farms in Canterbury, New Zealand, shows where bioassays revealed resistance in seedlings of one or more weed species.** Symbols whether the sample was from the random survey, or from ad-hoc reports of resistance. Resistant weeds included *A. fatua*, *B. catharticus*, *Lolium* spp., *P. minor*, *S. asper* and *S. oleraceus*. Base map and data from OpenStreetMap contributors and the OpenStreetMap Foundation.

*Lolium* plants collected near Timaru). We also documented resistance to ALS- and ACCase-inhibitors for *P. minor*, and to ALS-inhibiting herbicides in *B. catharticus* in two adjacent farms. Herbicide-resistance was found for more than one weed genus in 11 farms, and three of these farms had herbicide-resistant weeds from three genera. A total of 13 farms showed resistance to ALS- and ACCase-inhibiting herbicides (*A. fatua* for two farms) and (*Lolium* spp. for 12 farms)–two farms had resistance to both ALS- and ACCase-inhibitors for both *Lolium* spp. and *A. fatua* samples. Twelve randomly sampled farms (14%) had chlorsulfuron-resistant sow thistles, mostly *S. asper* but also *S. oleraceus*. We also tested the hypothesis that samples from farms in the Lincoln region were less likely to develop cases of herbicide resistance compared with Timaru because they are regarded as having more complex crop rotations. At Lincoln 22 out of 52 farms had herbicide resistance confirmed versus Timaru where it was 20 out of 35 farms. There was, however, no significant difference ($\chi$-squared = 1.2975, df = 1, p-value = 0.2547).

Industry supplied samples of *Lolium* spp. included cases of resistance to glyphosate (1 farm), ACCase-inhibitors (7 farms) and ALS-inhibitors (8 farms); those that occur in the survey areas have been included in the maps (Figs 1 & 2). From other parts of the country, we

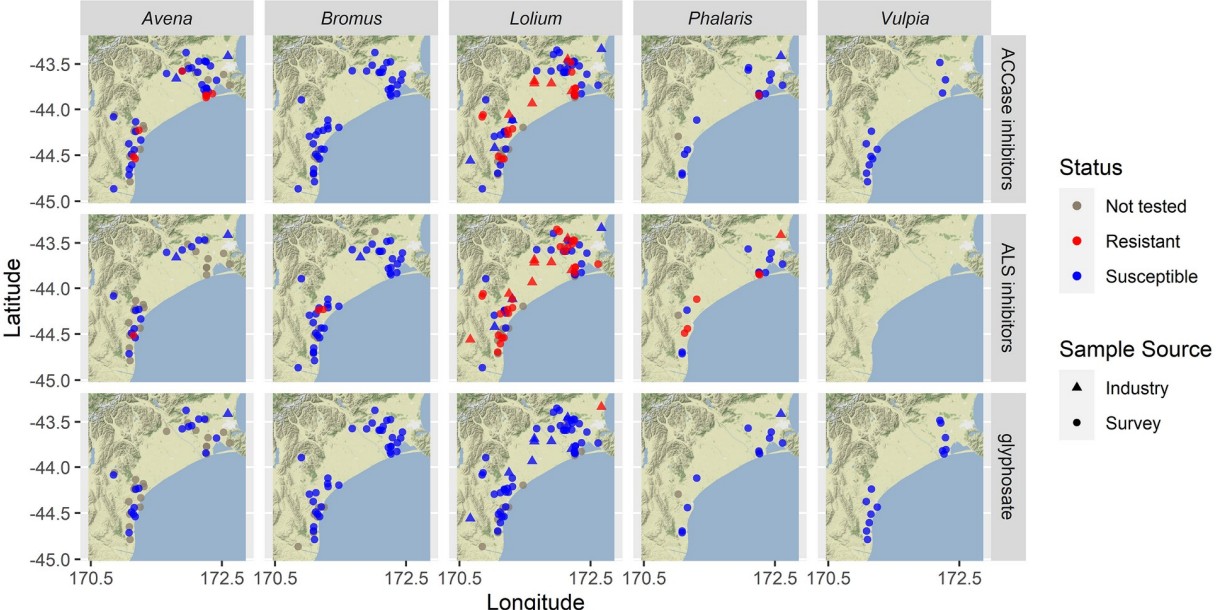

**Fig 2. Map of where weeds from each genus (vertical panels) survived or died.** Treatments involved using one or more herbicides in the herbicide groups indicated (horizontal panels); the specific herbicides used per species in each weed genus are mentioned in the Methods and supplementary data. Base map and data from OpenStreetMap contributors and the OpenStreetMap Foundation.

detected nicosulfuron-resistant *D. sanguinalis* from a farm in the Waikato region of the North Island (with 49–87% survival; S1 Appendix). We also had *S. media* samples sourced from ryegrass rotation (near Matamata in the North Island) and samples from Ashburton in the South Island (from barley and ryegrass crops) that showed resistance to flumetsulam (100% survival), and chlorsulfuron (95% survival).

## Discussion

This study is the first random survey carried out in New Zealand to detect herbicide resistance for a range of arable weeds and estimate its prevalence on wheat and barley farms. Such surveys may not have been implemented previously because costs of these investigations are prohibitive, an earlier estimate suggested it could cost as much as 759 NZD (New Zealand Dollars) per farm [27]. However, we estimated costs of approximately 370 NZD per farm in the second year of these surveys. After randomly sampling of >20% wheat and barley farms in the targeted regions, resistance was detected in 48% of the sampled farms, this is likely to be lower than the true rate since detection is imperfect [27]. The basis for this argument is that we could have missed individual resistant plants in a field and because we focused on up to ten plants in just two fields per farm, depending on which weeds were available to collectors prior to harvest [18,19]. Our sampling rate is better suited to the detection of outcrossing weed species but could miss some self-pollinating species [18,19]. Species previously identified as having an elevated risk of developing herbicide resistance in wheat and barley fields were confirmed resistant, i.e., *L. multiflorum*, *L. perenne*, *A. fatua*, *P. minor*, and *S. media* [6]. *Bromus catharticus* was resistant to ALS-inhibiting herbicides in this study, a first globally [5], but identified as medium to low risk by Ngow et al [6].

Before our survey was started, we believed there was a low prevalence of herbicide resistance in wheat and barley, for example, in our funding proposal for this work we estimated

that 5–10% of farms would contain resistant weeds. Only three previous publications documented cases of *Lolium* spp. or *A. fatua* resistant to ALS and ACCase herbicides in wheat and barley crops in New Zealand [11,12,28]. There was also a case of resistance of *S. media* in an oat crop (*Avena sativa* L.) [13]. However, we found that resistance is common overall (48%), and particularly for grass weeds on wheat and barley farms with plants surviving on between 23% and 36% of the farms after treatments with ACCase- and ALS-inhibiting herbicides, respectively. This suggests that this issue was historically under-reported by farmers, agricultural chemical suppliers, and consultants as well as under-investigated by scientists.

Because there was variation in effectiveness within herbicides that share the ACCase-inhibitors, this suggests different mutations could be involved [29]. For the ALS-inhibiting herbicides effective on grasses we did not try many of subclasses, e.g., only triazolopyrimidine (pyroxsulam), and sulfonylurea (iodosulfuron), so rates and types of cross-resistance are less clear. For the ACCase herbicides, some *A. fatua* oats survived fenoxaprop, but other herbicides within the same mode-of-action (clethodim, haloxyfop, and pinoxaden) remained effective where they were tested. Populations of *A. fatua* in Australia with resistance to fenoxaprop but not to other with ACCase-inhibiting herbicides, had a mutation at the Trp-1999-Cys site of the acetyl-CoA carboxylase coding region [30]. All *Lolium* spp. collected in the random survey died when treated with glyphosate. Clethodim was usually effective, but three farms had a population resistant to clethodim, pinoxaden and haloxyfop, perhaps implying an Ile-1781-Leu mutation or non-target site resistance [29]. We also documented the first New Zealand cases of *P. minor* surviving treatments with ACCase-inhibitors (fenoxaprop and clodinafop) and an ALS- inhibitor (pyroxsulam), and of *B. catharticus* surviving pyroxsulam. Other brome species in the United Kingdom are known to have developed both target and non-target site resistance to ALS-inhibitors [31]. In the case of *P. minor*, the herbicides haloxyfop, clethodim and glyphosate were effective for control of the populations resistant to pyroxsulam, fenoxaprop or clodinafop. Australia is also seeing *Sonchus* spp. with resistance to chlorsulfuron [32]. Rates of resistance in *Lolium* spp. are lower here than in *L. rigidum* populations from mainland Australia [33] but similar to rates seen in Tasmania [34].

Industry agronomists supplied us with *Lolium* spp. samples suspected of resistance in the field and we confirmed resistance to ALS-inhibitors, ACCase-inhibitors and glyphosate (Table 1). The latter case represented the first case of glyphosate resistant ryegrass plants sourced from a cereal crop (barley) in New Zealand, previous cases had been sourced from vineyards [35]. Other industry supplied samples of *S. media* led to us confirming resistance to flumetsulam and chlorsulfuron sourced from one farm in the Waikato and one farm in the Canterbury regions of New Zealand (in ryegrass and barley fields respectively). Similar resistance was reported from wheat fields in Canada [36]. Chlorsulfuron-resistant *S. media* from an oat field had been documented previously in Southland, New Zealand [13].

The herbicide rates we applied were the highest manufacturer recommended label rates for the herbicides registered in New Zealand. The label rates in New Zealand are often higher than those for other countries. For example, the highest recommended rates for controlling *L. multiflorum* in New Zealand is 1458 g ai/ha for glyphosate, 240 g ai/ha for clethodim, and 125 g ai/ha for haloxyfop, while the highest recommended rates on similar Australian product labels are 540 g ai/ha, 60 g ai/ha, and 54 g ai/ha, respectively. The discriminating doses used in other studies with similar systems, therefore, could be quite different [33,34]. Because of high *A. fatua* survival rates for the pyroxsulam at the recommended rate of 15 g ai/ha, we ended up respraying all our *A. fatua* populations at two times the label rate (30 g ai/ha), with the non-ionic adjuvant at 0.25% (linear alcohol ethoxylate 935 g/L); only one population was resistant at that rate. The Rexade GoDri® label in New Zealand suggests 15 g ai/ha rate is effective, but specifically in the presence of crop competition.

A few lessons were learned in the process of doing this work. In the first year of the survey, we bulked samples for multiple plants of the same species in a field, as described for survey work done in Australia [33]. The bulking of samples from a field was found to be a poor sampling strategy compared to sampling seeds from individual mother plants. We found that the cases of resistance were more obvious with samples from individual mother plants–for example, samples usually had >75% survival if they survived a treatment, which is expected because of the shared parentage of the seeds. If we mixed samples from multiple parents, the amount of resistant detected could vary and would depend on the proportion of plants in the field that were resistant. This work was focused on a binary question of whether there is resistance at the level of farms not prevalence within farms, but by keeping seeds from individual mother plants separate it would be possible to assess the proportion of collected mother plants that produce resistant progeny for any given farm. At Ruakura, in the second year of testing, we moved from planting out bulked samples into individual pots to planting into a single propagation tray with six lanes of seedlings planted out, where each lane contained seed from a different sample (from an individual mother plant) and this meant every tray could then have a susceptible population included.

Future work will include inheritance studies [37], genetic tests of target site [38,39] and non-target site [40] genes using weeds surviving treatments in this study and dose response tests for some cases. This should include tests of tillered plants to see if individuals in the resistant populations display resistance to multiple modes-of-action, as well as random surveys in other crop types, e.g., vineyards and maize. Another unaddressed question relates to determining if most herbicide-resistant weeds developed after herbicidal selection in New Zealand or if some may have developed overseas and been imported as seed contaminants [41].

## Supporting information

**S1 Appendix. Treatment (active ingredient) and survival by farm code and sample number, representing a detailed breakdown of the results presented in Table 2.**
(XLSX)

## Acknowledgments

First and foremost, we thank the farmers who gave us access to their farms; FAR (Foundation for Arable Research) staff Chelsea Dines, Alex Prince and Harry Washington for collecting the samples.

## Author Contributions

**Conceptualization:** Christopher E. Buddenhagen, Trevor K. James, M. Phil Rolston, Hossein Ghanizadeh.

**Data curation:** Christopher E. Buddenhagen, Zachary Ngow, Matilda Gunnarsson.

**Formal analysis:** Christopher E. Buddenhagen, Zachary Ngow, Hossein Ghanizadeh.

**Funding acquisition:** Trevor K. James.

**Investigation:** Christopher E. Buddenhagen, M. Phil Rolston, Matilda Gunnarsson, Fengshuo Li, Hossein Ghanizadeh.

**Methodology:** Christopher E. Buddenhagen, Zachary Ngow, Deborah L. Hackell, M. Phil Rolston, Richard J. Chynoweth, Matilda Gunnarsson, Fengshuo Li, Hossein Ghanizadeh.

**Project administration:** Christopher E. Buddenhagen, Trevor K. James, Matilda Gunnarsson.

**Resources:** Trevor K. James, Kerry C. Harrington.

**Supervision:** Christopher E. Buddenhagen, M. Phil Rolston, Hossein Ghanizadeh.

**Validation:** Christopher E. Buddenhagen, Trevor K. James, Richard J. Chynoweth, Matilda Gunnarsson, Kerry C. Harrington, Hossein Ghanizadeh.

**Visualization:** Christopher E. Buddenhagen.

**Writing – original draft:** Christopher E. Buddenhagen, Hossein Ghanizadeh.

**Writing – review & editing:** Christopher E. Buddenhagen, Trevor K. James, Zachary Ngow, Deborah L. Hackell, M. Phil Rolston, Richard J. Chynoweth, Kerry C. Harrington, Hossein Ghanizadeh.

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
