## [Decision Letter · Decision Letter 0]

19 Aug 2021

PONE-D-21-22929

Resistance to post-emergent herbicides is becoming common for grass weeds on New Zealand wheat and barley farms.

PLOS ONE

Dear Dr. Buddenhagen,

Thank you for submitting your manuscript to PLOS ONE. After careful consideration, we feel that it has merit but does not fully meet PLOS ONE’s publication criteria as it currently stands. Therefore, we invite you to submit a revised version of the manuscript that addresses the points raised during the review process.

There is a huge gap between two reviewers, But my personal experience on the subject shows me this manuscript meets with standards and no need for any genetics study. Indirect sentences should be preferred in the text although most part meets this. If you can explain varying numbers of sampled plants from each field, it will increase value of your paper. I have another suggestion also that you can use a confidence scale depending on the number of plants: 1-3 or 1-4 or 1-5. Less palnts sampled to more plants sampled. I think you can also statistically calculate this, please have consulted by  a statistician. You can do it without statistical remarks as well, but it will be criticised by readers..

Anyway, please submit your revised manuscript bya month. If you will need more time than this to complete your revisions, please reply to this message or contact the journal office at plosone@plos.org. Please include the following items when submitting your revised manuscript:

We look forward to receiving your revised manuscript.

Kind regards,

Ahmet Uludag, Ph.D.

Academic Editor

PLOS ONE

Journal Requirements:

4. We note that Figures 1 and 2 in your submission contain map images which may be copyrighted. All PLOS content is published under the Creative Commons Attribution License (CC BY 4.0), which means that the manuscript, images, and Supporting Information files will be freely available online, and any third party is permitted to access, download, copy, distribute, and use these materials in any way, even commercially, with proper attribution. For these reasons, we cannot publish previously copyrighted maps or satellite images created using proprietary data, such as Google software (Google Maps, Street View, and Earth). For more information, see our copyright guidelines: http://journals.plos.org/plosone/s/licenses-and-copyright.

a) You may seek permission from the original copyright holder of Figures 1 and 2 to publish the content specifically under the CC BY 4.0 license.  

Additional Editor Comments:

I think this manuscript is a good one although a reviewer rejected. Please, use indirect sentences as a referee pointed out. You maybe need to explain why you had only one sample and tell the readers it cannot create problem for assessing resistance distribution. I am not sure but maybe you can add confidence levels depending on the plants that you collected seeds. ! plant from a field can be considered confidence level 1, and with maximum level confidence level 4 and two more levels or you can make it less or more levels.

Reviewers' comments:

Reviewer's Responses to Questions

**Comments to the Author**

1. Is the manuscript technically sound, and do the data support the conclusions?

Reviewer #1: Yes

Reviewer #2: Partly

2. Has the statistical analysis been performed appropriately and rigorously? 

Reviewer #1: Yes

Reviewer #2: N/A

3. Have the authors made all data underlying the findings in their manuscript fully available?

Reviewer #1: Yes

Reviewer #2: Yes

4. Is the manuscript presented in an intelligible fashion and written in standard English?

Reviewer #1: Yes

Reviewer #2: No

5. Review Comments to the Author

Reviewer #1: Manuscript PONE-D-21-22929 is an elegant report of field surveys of weed populations resistant to ALS, ACCase and EPSPS inhibitor herbicides in wheat and barley fields in New Zealand. Buddenhage et al., in addition to confirming the herbicide resistance of different grassweeds, they also present the distribution maps of the main genus throughout New Zealand. The manuscript is well written, and although the work had its limitations, the authors have reiterated them throughout the text. The authors sampled 20% of the wheat and barley farms, which seems quite representative; and although the authors mention that the high incidence of resistance they found (48%) could be well above the real frequency, since they considered only visible plants before harvest (L318). However, what caught my attention is the number of plants collected per species collected to obtain seeds (1-10 plants, L105). In particular, I considered that this small sample size is not representative for each field, because according to Burgos (2015, 10.1614/WS-D-14-00019.1), although it is possible to work with seeds of 5-10 plants to cross-pollinating species, it is advisable to collect between 20-40 plants or 5000 seeds for self-pollinating species.

Otherwise, the manuscript has many typographical errors that are easily corrected with a careful review. Most of the small typographical errors are marked in the attached manuscript, but personally I suggest that the authors change all expressions made in second person (we, our) to third person.

Reviewer #2: Methodology inappropriate. Methodology should include genetic confirmation of resistance. Number of testing to low to confirm resistance. Poor results. Inappropriate writing style. Citations occasionally wrong.

6. PLOS authors have the option to publish the peer review history of their article (what does this mean?). If published, this will include your full peer review and any attached files.

Reviewer #1: No

Reviewer #2: No

---

## [Author Response · Author response to Decision Letter 0]

31 Aug 2021

1 September 2021

Dear PLOS,

Please consider our revised research article “Resistance to post-emergent herbicides is becoming common for grass weeds on New Zealand wheat and barley farms.” We thank the editor and reviewers for the helpful comments which guided us to make improvements to the manuscript.

We had a prior interaction with PLOS about this manuscript – the original manuscript was submitted with cover letter 14 April 2021, and we had a reply requesting revisions in an email from the editor on 20 August 2021.

The suggested edits were all implemented using track changes, the edits to the references/citations do not show in track changes in the normal way (they were done in a citation reference manager) so they are described separately below. I also included the copyright permissions for the OpenStreetMap per the instructions at https://journals.plos.org/plosone/s/figures. Itemized responses to the reviewers and to the revisions provided in a pdf of the original manuscript follow. I uploaded an excel version of the supplementary Appendix since PLOS one allows those files for the data. I think this means I do not need to add it to a repository. A csv was provided before if that is preferred.

Sincerely,

Chris Buddenhagen PhD

chris.buddenhagen (skype)

+64 221 04084 (cell)

Chris.Buddenhagen@agresearch.co.nz

 

Response to Reviewers

Reviewer comments

Additional Editor Comments: I think this manuscript is a good one although a reviewer rejected. Please, use indirect sentences as a referee pointed out. You maybe need to explain why you had only one sample and tell the readers it cannot create problem for assessing resistance distribution. I am not sure but maybe you can add confidence levels depending on the plants that you collected seeds. ! plant from a field can be considered confidence level 1, and with maximum level confidence level 4 and two more levels or you can make it less or more levels.

Response: 

Indirect sentences implemented.

Some sentences added about sample sizes (first paragraph of Materials and Methods). 

The supplemental data can be viewed by readers to see how many plants survived any treatment from any given farm. We chose a single threshold as a conservative measure of prevalence. We’ve added explanations in the text about our focus and added an uncertainty 95% confidence interval for the binomial estimate.: 

Lines 126-165: ” We focused on detecting the presence of resistant plants of any weed detected at the level of farms, not on within farm population level differences. Our sampling rates are more suited to the reliable detection of outcrossing species e.g. Lolium [18,19] , but the presence of each weed species within farms varied stochastically, and time and resource considerations came into play. Combined with our focus on just two fields, we accepted that our estimates of resistance prevalence in farms would be conservative (lower than the true rate).”

lines 529-544 “This work was focused on a binary question of whether there is resistance at the level of farms not prevalence within farms…” 

Reviewer #1: Manuscript PONE-D-21-22929 is an elegant report of field surveys of weed populations resistant to ALS, ACCase and EPSPS inhibitor herbicides in wheat and barley fields in New Zealand. Buddenhage et al., in addition to confirming the herbicide resistance of different grassweeds, they also present the distribution maps of the main genus throughout New Zealand. The manuscript is well written, and although the work had its limitations, the authors have reiterated them throughout the text. The authors sampled 20% of the wheat and barley farms, which seems quite representative; and although the authors mention that the high incidence of resistance they found (48%) could be well above the real frequency, since they considered only visible plants before harvest (L318). However, what caught my attention is the number of plants collected per species collected to obtain seeds (1-10 plants, L105). In particular, I considered that this small sample size is not representative for each field, because according to Burgos (2015, 10.1614/WS-D-14-00019.1), although it is possible to work with seeds of 5-10 plants to cross-pollinating species, it is advisable to collect between 20-40 plants or 5000 seeds for self-pollinating species.

Otherwise, the manuscript has many typographical errors that are easily corrected with a careful review. Most of the small typographical errors are marked in the attached manuscript, but personally I suggest that the authors change all expressions made in second person (we, our) to third person.

Response: The reviewer points out that we are aware of imperfect detection and the true rate of resistance could be higher than we estimated we’ve added the Burgos and mentioned the outcrossing issue. Our ability to find enough weeds for any given species was dependent on its abundance so this is not entirely in our control either.

Reviewer #2: Methodology inappropriate. Methodology should include genetic confirmation of resistance. Number of testing to low to confirm resistance. Poor results. Inappropriate writing style. Citations occasionally wrong.

Response: Genetic confirmation of resistance is a useful line of evidence. As the editor will know the methods work well for some target site mechanisms, but are not reliable for non-target site mechanisms. Our methods are classical bioassays widely used and should be acceptable as a means of identify cases with resistance for example see the Burgos and Panozzo article we cite.

I did not see any mention of specific problems with the citations but have gone through those to check them. See changes to the citations under a separate heading below. 

PDF edits

The edits are visible in the track changes version of the document, where specified line numbers refer to the lines on track changes document.

Indirect sentences added wherever “We” was highlighted.

Where appropriate we changed the Latin names to use an abbreviation for the genus (at second mention). We also added full Latin nomenclature for crops mentioned.

Line 14: Corresponding authors now includes email only.

Line 31-32: herbicide active ingredients are listed (this is all covered in Table 1, and is species specific while the abstract is framed generally).

Line 37: You asked that this be added to the abstract from the results section.

Lines 73-87 first paragraph of introduction: several typos e.g. species names were corrected.

Line 92: Reference 15 Hampton moved to end of sentence.

Lines 115-121: Indirect style implemented

Lines 125-165: Added sentences about the choices we made regarding the number of plants sampled. This included the addition of two Burgos references.

Lines 167-248: Typos, genus abbreviations and units litres to L as recommended.

Line 267 caption Table 1: bolded the title, reworded, added the adjuvant ingredients reference.

Table 1: adjuvant ingredients added at first mention in the table.

Line 253-260: indirect style

Lines 293-305: Statistical analysis. Copyright information for OpenStreetMaps added for map figures per instructions at https://journals.plos.org/plosone/s/figures and the OpenStreetMap website. Also binomial.test for confidence interval estimation.

Lines 307: First paragraph of results. Third person style implemented, crop taxa and genus abbreviations

Line 419: Table 2 added explanation for Site of action abbreviations.

Figure Captions: bolded first sentence, reworded and copyright wording added per recommendation see above.

Lines 363-409: The binomial 95% confidence interval was added for prevalence.

Lines 435: Discussion Reworded the sentence about our prevalence estimate and added the Burgos references [18-19]. Used genus abbreviations. 

Lines 529-544: Clarified the limitations of our conclusions in response to reviewer concerns about sample size, while also outlining our change to sampling seed from individual mother plants instead of bulking samples.

Lines 545-552: Indirect style. Addition of a citation to work that raises the possibility that some resistant weeds are arriving in New Zealand from seed contaminants – a new line of investigation that we are pursuing.

Reference changes

[2] Stephenson – change case of title

[3] Thorne et al. – added missing page numbers and issue

[5] Heap – changed date accessed

[14] Ghanizadeh & Harrington – add issue, year and page numbers (I was citing early online version).

[15] Hampton et al – added missing page numbers and issue

[16] Busi et al – added page numbers and issue

[17] Holden – added new reference to NZ agrichemical manual

[18] Burgos – new reference

[19] Burgos et al – new reference

[20] Panozzo et al – added issue/volume change case

[22] Ghanizadeh & Harrington – add pages volume date

[23] Kahle & Wickam – change case

[27] Buddenhagen et al – page numbers date and volume

[33] Boutsalis et al – change case

[36] Laforest et al – change case of title

[37] Ghanizadeh et al – change case of title

[39] Malone et al – italics for species.

[40] Ghanizadeh & Harrington – change case of title.

[41] Rubenstein et al – new reference.

---

## [Decision Letter · Decision Letter 1]

17 Sep 2021

PONE-D-21-22929R1Resistance to post-emergent herbicides is becoming common for grass weeds on New Zealand wheat and barley farms.PLOS ONE

Dear Dr. Buddebhagen,

Thank you for submitting your manuscript to PLOS ONE. After careful consideration, we feel that it has merit but does not fully meet PLOS ONE’s publication criteria as it currently stands. Therefore, we invite you to submit a revised version of the manuscript that addresses the points raised during the review process.

 Could you please give attention to typos. I suggest please avoid wording such as resistant farms. I hope there will be no need for further correction after your new version.

We look forward to receiving your revised manuscript.

Kind regards,

Ahmet Uludag, Ph.D.

Academic Editor

PLOS ONE

Journal Requirements:

Additional Editor Comments:

It is almost done. As you will see reviewers' comments you need to give more attention to writing.

Reviewers' comments:

Reviewer's Responses to Questions

**Comments to the Author**

1. If the authors have adequately addressed your comments raised in a previous round of review and you feel that this manuscript is now acceptable for publication, you may indicate that here to bypass the “Comments to the Author” section, enter your conflict of interest statement in the “Confidential to Editor” section, and submit your "Accept" recommendation.

Reviewer #1: All comments have been addressed

Reviewer #3: All comments have been addressed

2. Is the manuscript technically sound, and do the data support the conclusions?

Reviewer #1: (No Response)

Reviewer #3: Partly

3. Has the statistical analysis been performed appropriately and rigorously? 

Reviewer #1: Yes

Reviewer #3: Yes

4. Have the authors made all data underlying the findings in their manuscript fully available?

Reviewer #1: Yes

Reviewer #3: Yes

5. Is the manuscript presented in an intelligible fashion and written in standard English?

Reviewer #1: No

Reviewer #3: Yes

6. Review Comments to the Author

Reviewer #1: The authors have satisfactorily addressed most of the comments in the previous version and intellectually the manuscript is acceptable for publication; however, I still found various typos throughout the manuscript.

L33: Resistant farms????

Maybe: Weeds, such as Avena fatua (9%, 1%, 0% of farms), Bromus catharticus (0%, 2%, 0%), Lolium spp. (17%, 28%, 0%), Phalaris minor (1%, 6%, 0%), and Vulpia romoides (0%, not tested, 0%), from different farms (denominator is 87 farms) displayed resistance to ACCase-inhibitors, ALS-inhibitors, and glyphosate

L38: change Phalaris to P.

L39: Bromus to B.

L41: Sonchus asper but also S. oleraceus.

L41: Check double point

L43: Phalaris to P.

L84: [17]space(compared

L116: Lolium multiflorum (when the scientific name begins the phrase, the genus is spelled out in full)

L138: Canterbury).

L156, L193 and L286: Avena fatua

L181: Sonchus asper and S. oleraceus (if the mention of species has no hierarchy, they are listed alphabetically)

L211-213: abbreviate the genus of the species

L266 and L341: list the scientific names alphabetically

L341: Bromus …

Reviewer #3: This study is an important and intensive survey for resistance weed species but it is a kind of preliminary study of finding suspicious herbicide resistance biotypes. Since the sample size and treatments is not enough, the conclusions should be reconsidered to mention herbicide resistance. Minor revisions were made in the text.

7. PLOS authors have the option to publish the peer review history of their article (what does this mean?). If published, this will include your full peer review and any attached files.

Reviewer #1: **Yes: **Ricardo Alcántara-de la Cruz

Reviewer #3: **Yes: **Filiz ERBAS

---

## [Author Response · Author response to Decision Letter 1]

21 Sep 2021

21 September 2021

Dear PLOS,

Please consider our revised research article “Resistance to post-emergent herbicides is becoming common for grass weeds on New Zealand wheat and barley farms.” We thank the editor and reviewers for the helpful comments which guided us to make improvements to the manuscript.

We had a prior interaction with PLOS about this manuscript – the original manuscript was submitted with cover letter 14 April 2021, and we had a reply requesting revisions in an email from the editor on 20 August 2021, followed by a decision: revision required PONE-D-21-22929R1 on 18 September 2021.

Most edits made using track changes by the reviewers accepted. Other suggestions were implemented unless indicated otherwise. Itemized responses to the reviewers comments follow in this letter. 

Sincerely,

Chris Buddenhagen PhD

chris.buddenhagen (skype)

+64 221 04084 (cell)

Chris.Buddenhagen@agresearch.co.nz

 

Response to Reviewers

Reviewer comments

Editor Comments: Could you please give attention to typos. I suggest please avoid wording such as resistant farms. - Done

Response: 

Reviewer #1: The authors have satisfactorily addressed most of the comments in the previous version and intellectually the manuscript is acceptable for publication; however, I still found various typos throughout the manuscript.

L28: Glyphosate’s mode of action can be added. – Although there is only one herbicide with this mode of action, we added it as requested by a reviewer. It seems unnecessary.

L33: Resistant farms???? – reworded to indicate the weeds are resistant

Maybe: Weeds, such as Avena fatua (9%, 1%, 0% of farms), Bromus catharticus (0%, 2%, 0%), Lolium spp. (17%, 28%, 0%), Phalaris minor (1%, 6%, 0%), and Vulpia romoides (0%, not tested, 0%), from different farms (denominator is 87 farms) displayed resistance to ACCase-inhibitors, ALS-inhibitors, and glyphosate – reworded to indicate the weeds are resistant

L38: change Phalaris to P. – changed

L39: Bromus to B. – changed

L41: Sonchus asper but also S. oleraceus. – changed

L41: Check double point – changed

L43: Phalaris to P.v– changed

L84: [17]space(compared – changed

L84: Reviewer comment: Since the sentence, starting with “Another plausible expalanation…” gives a reason for low number of resistance. It is understood that higher application rates is one of reason for low resistance cases. It can not be the reason for low number of resistance. I couldn’t see any discussion about this sentence at Discussion part below, either.

Response:

Herbicides applied at registered rates can clearly select for major gene (e.g., target-site) resistance, whereas initially, suboptimal herbicide rates may select for both major and minor gene (i.e. quantitative) resistance. We discuss the difference in rates between Australia and NZ later. We would prefer to leave this sentence in the article.

L116: Lolium multiflorum (when the scientific name begins the phrase, the genus is spelled out in full) – changed

L138: Canterbury). – changed

L156, L193 and L286: Avena fatua – changed

L163: Reviewer comment: These sentences can be converted into a table with information on the name of these weed species, sample size, where it was collected, which herbicide or mode of action it was tested against

Response:

We would like to leave this here- it draws a distinction between the samples processed in the random survey and the ones that were supplied as suspected cases of resistance by industry agronomists, and it includes well as one broadleaf weed that we tested. The grasses are also in Table 2 and all are in the Appendix S1.

159 – moved the sentences about susceptible controls to the “Collection of plant material” section per recommendation.

L181: Sonchus asper and S. oleraceus (if the mention of species has no hierarchy, they are listed alphabetically) – changed [though this is not a rule I have heard before]

L211-213: abbreviate the genus of the species – changed

L266 and L341: list the scientific names alphabetically – changed

L341: Bromus …– changed

L269: Reviewer comment: In order to be able to talk about herbicide resistance, discriminating doses and label doses should be used for each active substance and each population in pot studies. I have seen that you used more than one dose for some weeds. But to make confirmation of herbicide resistance for all the weeds you studied, more comprehensive studies must be performed. So instead of using certain statement of herbicide resistance, suspicion of herbicide resistance should be mentioned for weeds and active ingredients. This study is a kind of preliminary study of finding suspicious herbicide resistance biotypes.

Also in the email from the editor this comment is similar:

Reviewer #3 comment: This study is an important and intensive survey for resistance weed species but it is a kind of preliminary study of finding suspicious herbicide resistance biotypes. Since the sample size and treatments is not enough, the conclusions should be reconsidered to mention herbicide resistance. Minor revisions were made in the text. 

Response:

By the definition the recommended dose is a discriminating dose, it is the dose the manufacturer has determined to kill the weeds on the label. When we detect resistant plants there is every reason to believe they are resistant. We experimentally confirmed the resistance by spraying with this dose and it did not kill the resistant plants indicated here, but it did kill susceptible reference populations. We are not the first to use the label rate as a discriminating dose: Broster et al 2012 cited, and the following references. 

Broster JC, Koetz EA, Wu H 2010. A survey of southern New South Wales to determine the level of herbicide resistance in brome grass and barley grass populations. In: Zydenbos SM ed. Christchurch, New Zealand. Seventeenth Australasian Weeds Conference: 274–277.

Broster JC, Koetz EA, Wu H 2011. Herbicide resistance levels in annual ryegrass (Lolium rigidum Gaud.) in southern New South Wales. Plant Protection Quarterly 26: 22–28.

Owen MJ, Martinez NJ, Powles SB 2014. Multiple herbicide-resistant Lolium rigidum (annual ryegrass) now dominates across the Western Australian grain belt. In: Iannetta P ed. Weed Research 54: 314–324.

Owen MJ, Martinez NJ, Powles SB 2015a. Herbicide resistance in Bromus and Hordeum spp. in the Western Australian grain belt. Crop and Pasture Science 66: 466.

We responded to the idea that the sample size is inadequate in the last peer review by indicating that our estimate of the number of farms with resistant plants is likely conservative. Bear in mind that even with much higher sampling rates, you would still be unable to determine if you detected all cases of resistance that might be present. We do not exaggerate our claims about resistance rates.

We state this: 

In the methods we state this:

Our sampling rates are more suited to the reliable detection of outcrossing species e.g., Lolium [18,19], but the presence of each weed species within farms varied stochastically, and time and resource considerations came into play. Combined with our focus on just two fields, we accepted that our estimates of resistance prevalence in farms would be conservative (lower than the true rate).

In the discussion:

After randomly sampling of >20% wheat and barley farms in the targeted regions, resistance was detected in 48% of farms, this is likely to be lower than the true rate since detection is imperfect [27]. The basis for this argument is that we could have missed individual resistant plants in a field and because we focused on up to ten plants in just two fields per farm, depending on which weeds were available to collectors prior to harvest [18,19]. 

We added this in the discussion this time: Our sampling rate is better suited to the detection of outcrossing weed species but could miss some self-pollinating species [18.19].

L379: Reviewer comment: From the beginning I don’t understand whay you tested glyphosate while you are surveying cereal farms, since glyphosate is not recommended to use in cereals. You can add some expalanatory sentences why you choosed glyphosate. For example; plant rotation or resistance seed dispersal.

Response:

Glyphosate is commonly used for seed bed preparation and glyphosate resistant ryegrass is known from elsewhere in New Zealand. Under the section growing plants and spraying the following sentence was added: “Glyphosate was tested because it is commonly used prior to planting for seed bed preparation.” In addition the detection of glyphosate resistant ryegrass in a barley crop proves that we were right to look at this herbicide.

---

## [Decision Letter · Decision Letter 2]

4 Oct 2021

Resistance to post-emergent herbicides is becoming common for grass weeds on New Zealand wheat and barley farms.

PONE-D-21-22929R2

Dear Dr. Buddenhagen,

We’re pleased to inform you that your manuscript has been judged scientifically suitable for publication and will be formally accepted for publication once it meets all outstanding technical requirements.

Kind regards,

Ahmet Uludag, Ph.D.

Academic Editor

PLOS ONE

Additional Editor Comments (optional):

Congratulations.

Reviewers' comments:

Reviewer's Responses to Questions

**Comments to the Author**

1. If the authors have adequately addressed your comments raised in a previous round of review and you feel that this manuscript is now acceptable for publication, you may indicate that here to bypass the “Comments to the Author” section, enter your conflict of interest statement in the “Confidential to Editor” section, and submit your "Accept" recommendation.

Reviewer #1: All comments have been addressed

Reviewer #3: All comments have been addressed

2. Is the manuscript technically sound, and do the data support the conclusions?

Reviewer #1: Yes

Reviewer #3: Yes

3. Has the statistical analysis been performed appropriately and rigorously? 

Reviewer #1: Yes

Reviewer #3: Yes

4. Have the authors made all data underlying the findings in their manuscript fully available?

Reviewer #1: Yes

Reviewer #3: Yes

5. Is the manuscript presented in an intelligible fashion and written in standard English?

Reviewer #1: Yes

Reviewer #3: Yes

6. Review Comments to the Author

Reviewer #1: (No Response)

Reviewer #3: All the comments have been answered properly. Manuscript is technically sound and written in standart English. Statistical analysis has been performed aprropriately and the authors made all the data available.

7. PLOS authors have the option to publish the peer review history of their article (what does this mean?). If published, this will include your full peer review and any attached files.

Reviewer #1: **Yes: **Ricardo Alcántara-de la Cruz

Reviewer #3: **Yes: **Filiz ERBAS

---

## [Editor Report · Acceptance letter]

6 Oct 2021

PONE-D-21-22929R2 

Resistance to post-emergent herbicides is becoming common for grass weeds on New Zealand wheat and barley farms. 

Dear Dr. Buddenhagen:

I'm pleased to inform you that your manuscript has been deemed suitable for publication in PLOS ONE. Congratulations! Your manuscript is now with our production department. 

Kind regards, 

on behalf of

Dr. Ahmet Uludag 

Academic Editor

PLOS ONE